# De-Anonymizing Text by Fingerprinting Language Generation

**Zhen Sun**
Cornell University
zs352@cornell.edu

**Roei Schuster**
Cornell Tech, Tel Aviv University
rs864@cornell.edu

**Vitaly Shmatikov**
Cornell Tech
shmat@cs.cornell.edu

## Abstract

Components of machine learning systems are not (yet) perceived as security hotspots. Secure coding practices, such as ensuring that no execution paths depend on confidential inputs, have not yet been adopted by ML developers. We initiate the study of code security of ML systems by investigating how nucleus sampling—a popular approach for generating text, used for applications such as auto-completion—unwittingly leaks texts typed by users. Our main result is that the series of nucleus sizes for many natural English word sequences is a unique *fingerprint*. We then show how an attacker can infer typed text by measuring these fingerprints via a suitable side channel (e.g., cache access times), explain how this attack could help de-anonymize anonymous texts, and discuss defenses.

## 1 Introduction

Machine learning (ML) models are composed from building blocks such as layer types, loss functions, sampling methods, etc. Each building block typically has a few popular library implementations, which are incorporated into many models—including models whose inputs are sensitive (e.g., private images or typed text). Therefore, ML models are "security hotspots" and their implementations must follow secure coding practices. This includes protecting the inputs from *side channels*, i.e., low-level physical or microarchitectural side effects of the computation that are externally observable and leak information about its internal state to concurrent, adversarial processes.

We use *nucleus sampling* [19], a leading approach for efficiently generating high-fidelity text, as a case study of side-channel vulnerabilities in ML models. Given the output probabilities of a language model such as GPT-2 [35], nucleus sampling draws candidates from a variable-sized "nucleus" of the most probable words. It is the basis of applications such as text auto-completion [24, 44].

First, we demonstrate that **the series of nucleus sizes produced when generating an English-language word sequence is a fingerprint** by showing that the nucleus size series of any sequence satisfying a simple criterion is far from any other sequence, unless their textual contents substantially overlap. We then derive a lower bound on the Euclidean distance between fingerprints that depends only on the sequence length but not on the size or domain of the corpus.

Second, we show that implementations of nucleus sampling, such as the popular Hugging Face Transformers package, contain a dangerous information leak. An attacker who runs a concurrent, sandboxed application process on the user's device can infer the nucleus size by indirectly measure the number of iterations of a certain loop, and thus fingerprint the input text. We use Flush+Reload [47] for our proof of concept, but the general approach works with any suitable side channel [17, 28, 32].

We design a fingerprint matching algorithm and show that (1) it tolerates noise in side-channel measurements, and (2) does not produce false positives. Therefore, an attacker can accurately identify the typed sequence out of many billions of possible candidates in an "open-world" setting, without assuming *a priori* that the user's input belongs to a known small dataset. This technique can help

de-anonymize text by asynchronously matching fingerprints collected from the user's device to anonymous blog entries, forum posts, emails, etc. For example, we show that many of the anonymous users' posts on the infamous Silk Road forum have unique fingerprints.

We conclude by explaining how to mitigate the information leak and discuss the importance of removing insecure coding patterns such as input-dependent loops from ML building blocks.

***Ethics and responsible disclosure.*** The purpose of this study is to improve the security of popular ML systems and help protect the privacy of their users. We disclosed our findings and our proposed mitigation code by email to members of the Hugging Face engineering team responsible for the implementation of nucleus sampling (identified via a contact at Hugging Face and GitHub's commit log) and a message to Hugging Face's public Facebook contact point.

We use Silk Road posts as a case study only because they represent informal textual communications whose authors likely wish to maintain their anonymity. **Silk Road posts include offensive and harmful content. We use this dataset solely for our proof-of-concept experiments. It does not reflect our views in any way.**

## 2 Background

### 2.1 Text generation via language model sampling

Let $\mathbb{D}$ be a dictionary, $\mathbb{S} = \cup_{i \in \mathbb{N}} \mathbb{D}^i$ a set of possible *texts* (sequences of dictionary words), and $X \in \mathbb{S}$. A *language model* $\mathcal{M} : \mathbb{S} \to \mathbb{R}^{|\mathbb{D}|}$ maps a "prefix" $(x_1, \ldots, x_n) \in \mathbb{S}$ to a probability distribution $(p_1, \ldots p_{|\mathbb{D}|})$ of the next word. **Text auto-completion** is a popular application of language generation. As the user is typing some text $X \in \mathbb{S}$, a language model is sampled at each time step $t \in \{1, .., |X|\}$, to generate a "probable" suffix for $X[:t]$ (the prefix of $X$ up to index $t$).

*Pure* sampling draws the next word $y$ according to the probabilities given by $\mathcal{M}(x_1, \ldots x_n)$, then invokes $\mathcal{M}$ on $(x_1, \ldots, x_n, y)$, and so on. Typically, sampling stops when a special end-of-sequence or end-of-sentence token is sampled, or when the probability of the entire sampled sequence (estimated by multiplying the model's output probabilities for the sampled words) drops below a certain threshold. Other approaches include *greedy* sampling, which simply sets $x_{n+1} \leftarrow \operatorname{argmax} \mathcal{M}(x_1, \ldots, x_n)$, and *top-k* sampling, which selects words corresponding to the top $k$ highest values in $\mathcal{M}(x_1, \ldots x_n)$ and applies pure sampling to them according to their probabilities (normalized to sum up to 1). Different sampling methods generate text with different properties [19, 43]. Pure sampling produces poor-quality text (often, incomprehensible gibberish) as perceived by humans, while greedy sampling results in a lack of language diversity, often with highly unnatural repetition.

**Nucleus sampling** [19] is similar to top-k sampling but instead of choosing candidates based on ranks, it chooses the maximal set ("nucleus") of top-ranked words such that the sum of their probabilities is $\leq q$. It produces high-quality, high-diversity text [19] and performs well on metrics, including the Human Unified with Statistical Evaluation (HUSE) score [18].

### 2.2 Microarchitectural side channels

Process isolation in modern systems is a leaky abstraction. If a user's process and an attacker's concurrent process share physical hardware resources, the attacker can infer information about the user's activity by analyzing contention patterns on the cache (see below), cache directories [45], GPU [30], translation lookaside buffer [15], and many other resources. These attacks, known as *microarchitectural side channels*, can be exploited by any untrusted, low-privilege process. Side-channel attacks have been demonstrated on many PC and mobile [28] platforms, and even from Javascript or WebAssembly code within the highly restricted browser sandbox [14, 31].

Several programming patterns are especially vulnerable to side-channel attacks. Loop arguments are a textbook example [27]: loops take longer to execute than non-iterative code, their execution time can be inferred using coarse timers, and their side effects on microarchitectural resources are repeated many times, amplifying the signal. Loops whose iterations depend on some secret can leak this secret through many microarchitectural [15, 29, 47] and physical [12, 13] side channels.

***Cache side channels.*** Cache memory is shared even among isolated processes. When the contents of physical memory addresses are loaded or evicted by any process, it affects how fast that memory

can be accessed by other processes. Therefore, memory access times measured by one process can reveal which memory addresses are accessed by another process. Cache attacks have been used to extract cryptographic keys [7, 29, 32, 33, 47, 49], steal login tokens [37], defeat OS security mechanisms [25], sniff user inputs [28], and more [36, 49, 50].

*Flush+Reload* [17, 47] is a popular type of cache attacks. When a victim process and a concurrent attacker process load the same shared library or a file, a single set of physical memory addresses containing the file's content is mapped into both processes' virtual address space. In this situation, the attacker can (1) cause the eviction of a specific memory address ("flush"), (2) wait, and (3) reload this memory address. Short reload time reveals that the victim has accessed this address after the eviction but before the load. If memory addresses monitored by the attacker do not contain shared memory, other cache attacks such as Prime+Probe [32, 33] may be used instead.

## 3 Fingerprinting auto-completed text sequences

Consider a text auto-completion assistant that uses nucleus sampling (see Section 2.1). At each step $t$, the user has typed $X[:t]$. The assistant uses $\mathcal{M}(X[:t])$ to generate nucleus $q$ and samples from it to auto-complete the user's text. The user may accept the completion or manually type in the next word. For texts that are typed over multiple "sessions," e.g., in separate forum posts, we assume the assistant stops sampling at the end of every text and resets. Let $\mathcal{I}_{\mathcal{M},q}(X) \in \mathbb{R}^{|X|}$ be the resulting *nucleus size series* (NSS). Figure 1 shows an example.

### 3.1 Fingerprints of text sequences

Let $X, Y \in \mathbb{S}$ be text sequences s.t. $|X| = |Y|$. We say that $X$ and $Y$ are *similar* if they have identical subsequences of length $N$ starting at the same index, i.e., $\exists i \in \mathbb{Z}, 0 \le i < |X| - N$ s.t. $X[i:i+N] = Y[i:i+N]$. We set $N = 50$, which is a very rigorous criterion for similarity: if two sequences have a common 50-word subsequence in exactly the same position, they are likely identical or have some identical source (and are thus *semantically* close to each other).

Let $\pi$ be a procedure that receives as input $X \in \mathbb{S}$ and returns a vector in $\mathbb{R}^{|X|}$. We say that $\pi(X)$ is a **fingerprint** if there exists a monotonically increasing *uniqueness radius* $U : \mathbb{N} \to \mathbb{R}$ such that for any $Y \in \mathbb{S}$ which is not similar to $X$ and $|Y| = |X|$, $\|\pi(X) - \pi(Y)\| > U(|X|)$ where $\|.\|$ is the Euclidean norm. In other words, there exists a "ball" around the fingerprint of any sequence $X$ such that no other sequence has its fingerprint within that ball (unless it is similar to $X$). This defines an *open-world* fingerprint, i.e., the uniqueness of a sequence's fingerprint holds with respect to all natural-language sequences and not just a specific dataset.

### 3.2 Nucleus size series is a fingerprint

We conjecture that $\pi(X) = \mathcal{I}_{\mathcal{M},q}(X)$ of any English sequence is a fingerprint, as long as $\mathcal{I}_{\mathcal{M},q}(X)$ is sufficiently "variable." We define variability of $\mathcal{I}_{\mathcal{M},q}(X) = (n_1, \ldots, n_{|X|})$ as $\sqrt{\frac{1}{|X|} \sum_{i=1}^{|X|} (n_i - \mu)^2}$, where $\mu = \frac{1}{|X|} \sum_{i=1}^{|X|} n_i$ (by analogy with statistical variance). We say that $X$ is *variable* if variability of its NSS is greater than some $T \in \mathbb{R}$. $T$ depends on the language model $\mathcal{M}$, and is set to 1450 in our experiments.

It is computationally infeasible to compute $\|\pi(X) - \pi(Y)\|$ for every pair $X, Y \in \mathbb{S}$ in the English language. To validate our conjecture, we show that when a "variable" $X$ and another sequence $Y$ are sampled from a real-world English corpus and $X$ and $Y$ are not similar, it always holds that $\|\pi(X) - \pi(Y)\| > U(|X|)$ for a large $U(|X|)$. Critically, $U$ depends only on the sequence length but not the size or domain of the corpus from which $X$ is drawn. Furthermore, this holds for any $Y$, variable or not. This implies that there are no other fingerprints within the $U$-radius ball of $\pi(X)$.

***Generating NSS.*** We downloaded 5 "subreddit" archives from Convokit [8] that have the fewest common users: asoiaf, india, OkCupid, electronic_cigarette, and Random_Acts_of_Amazon. We also downloaded the sports subreddit that has more active users and posts. We then aggregated each user's posts into longer sequences (up to 3000 words) by concatenating them in chronological order.

To simulate auto-completion running in the background while a text sequence is being typed, we invoke the Hugging Face Transformers language generator (`run_generation.py`) to drive a GPT-2 [35] language model (`gpt2-small`) and output one word for every prefix. We use nucleus size

with $q = 0.9$. To reduce computational complexity, we modified the script to save the encoder's hidden state for every prefix, so it is necessary to decode only one additional word for the next prefix.

***Many sequences are variable.*** The fraction of variable sequences depends on the domain: 43.6% for the OkCupid dataset, 62.8% for asoiaf, 77.6% for india, 71.6% for electronic_cigarette, 41.6% for Random_Acts_Of_Amazon, 42% for sports. Variability seems to be strongly and inversely correlated with the average post length, i.e., short posts result in high variability. The average post length is 50.3 for OkCupid dataset, 35.8 for asoiaf, 38.0 for india, 42.9 for electronic_cigarette, 49.2 for Random_Acts_Of_Amazon, 60.6 for sports. We conjecture that the (re-initialized) language model is more "uncertain" about the next word at the beginning of posts, and large nucleus sizes correspond to high variability. Figure 1 illustrates this effect.

We conclude that, even when posts are relatively long (e.g., sports), many (>40%) sequences are variable. This fraction may be lower when individual texts are much longer, e.g., blog posts.

***NSS of a variable sequence is unique.*** We measure pairwise Euclidean distances between the NSS of variable sequences and the NSS of other (not necessarily variable) sequences. Figure 2a shows the histogram (smoothed by averaging over a 10-bucket window) for 500 randomly chosen 2700-word sequences from the OkCupid dataset. Sample density decreases exponentially with distance from the density peak, which is around 105k. Because we omit the pairs where neither NSS is variable, this effect is asymmetric: density decreases slower above the peak than below the peak. Exponential decay to the left ensures that the lowest values observed in practice are never too far from the peak.

To verify this on a larger scale, we confirmed that the lowest pairwise distance between a variable NSS and any other NSS is consistent regardless of the dataset size (Figure 2b) or domain (Figure 2c).

**Algorithm 1** Nucleus sampling [24]

```
1: procedure SAMPLE_SEQUENCE(M, p, X)
2:     logits ← M(X) // get logits
3:     logits ← TOP_P_FILTERING(logits, p)
4:     next ← MULTINOMIAL_SAMPLE(logits)
5:     return next
6:
7: procedure TOP_P_FILTERING(logits, p)
8:     sorted_logits, indices ← DESCEND_ARGSORT(logits)
9:     cum_probs ← CUM_SUM(SOFTMAX(sorted_logits))
10:    not_in_p ← [ ]
11:    for i ∈ {1...LEN(logits)} do
12:        if cum_probs[i] > p then
13:            not_in_p.APPEND(indices[i])
14:    for i ∈ not_in_p do        ← number of iterations
15:        logits[i] ← −∞            corresponds to nucleus size
       return logits
16:
```

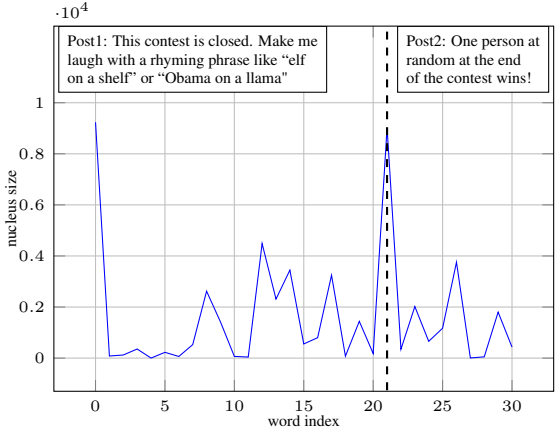

Figure 1: Concatenated NSS of 2 posts.

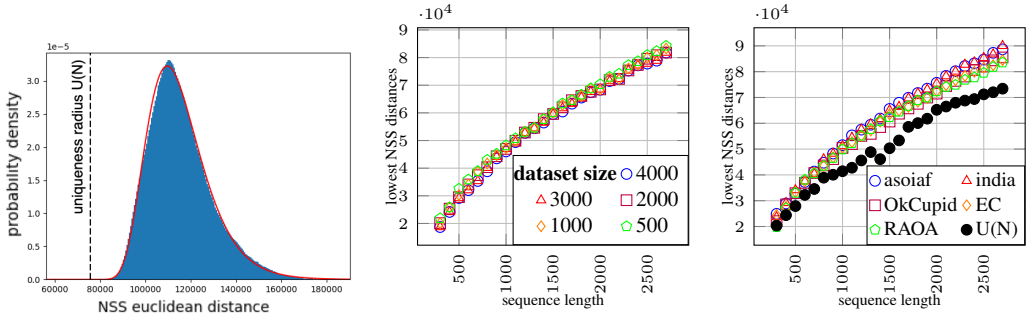

(a) NSS pairwise distances for 500 random 2700-word sequences (OkCupid).

(b) Lowest NSS distances for different dataset sizes (sports).

(c) Lowest NSS distances for different domains.

Figure 2: Nucleus size series (NSS) are fingerprints.

***NSS of a variable sequence is a fingerprint.*** To formally satisfy our definition of a fingerprint, NSS of variable sequences must have a uniqueness radius that depends on $N$. To show this for a given

$N$, we take the dataset with the lowest average variability and fit a log-normal distribution, which is the best fit among 90 distributions [10], to its sample histogram, as shown in Figure 2a. We chose $U(N)$ such that, on our fitted distribution, the probability to sample an element lower than $U(N)$ is $\epsilon \equiv 10^{-18}$, which we consider negligible. Figure 2c shows $U(N)$ for various $N$.

### 3.3 Execution path of nucleus sampling reveals nucleus sizes

Algorithm 1 shows the pseudo-code of nucleus sampling. After obtaining the probability of each possible next token from the language model, it calls TOP_P_FILTERING, which sorts and sums up the probabilities. It then selects the tokens whose cumulative probability is outside $top\_p$ and sets the corresponding logits to $-\infty$, i.e., removes these tokens. If an adversary can infer the number of loop iterations in line 14, he can learn the number of tokens removed from the vocabulary and thus the nucleus size, which is equal to the vocabulary size minus the number of removed tokens.

Auto-completion exposes not just the nucleus size at each step, but also the number of completed words before it stops due to low probability or end-of-sequence token. The series of these numbers, in addition to nucleus sizes, may be an even stronger fingerprint, but we leave this to future work.

## 4 Attack overview

### 4.1 Threat model

Consider a user who types text into a program that uses an auto-completion assistant based on nucleus sampling. At the same time, an attacker is running a concurrent, low-privilege process on the user's machine, (e.g., inside another application). Memory isolation and sandboxing ensure that the attacker's process cannot directly access the user's keyboard entries, nor the resulting text.

The attacker's goal is to infer the nucleus size series, which are revealed through the loop iteration count (see Section 3.3), via any available side channel (see Section 2.2). We assume that the attacker has access to the victim's language-model implementation. This is plausible for popular, publicly released code as GPT-2 and Hugging Face. If the victim is using an off-the-shelf auto-completion assistant as part of a commercial software package, the same code is likely available to potential attackers. Therefore, for any candidate text, the attacker can (re)produce the corresponding nucleus size series by re-running the model on the prefixes of this text.

We also assume that the attacker's measurement of the side channel is "aligned," i.e., the attacker can tell when the language model is queried to auto-complete a prefix (inferring this is relatively straightforward—see Section 5.1). The measurement can be imprecise, but we show that the error is bounded (see Appendix A in supplementary materials).

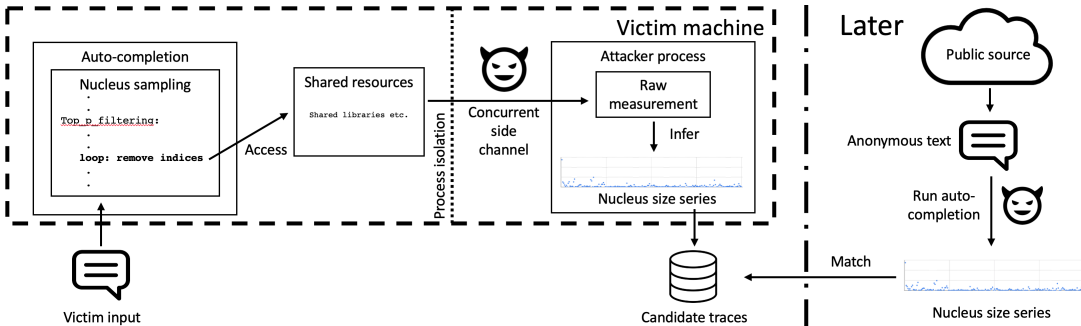

Figure 3: Attack overview.

One application of this attack is de-anonymization (see Figure 3). Consider a user who anonymously publishes some text on Reddit, Twitter, or a blog. In the *online phase* of the attack, while the user is typing, the attacker collects a trace by measuring the available side channel. The attacker stores all traces, along with user identifiers such as the IP address of the machine where the trace was collected. In a later, *offline phase*, the attacker obtains anonymously published texts and attempts to match them against the collected traces.

## 4.2 Matching an anonymous text to a side-channel trace

Algorithm 2 shows how the attacker can match a text sequence $X$ against previously collected traces. First, generate the nucleus size series (NSS) of $X$. If this NSS is sufficiently long and variable (see Section 3.2), GENTRACES($candidate\_traces, |X|$) creates a list of candidate traces whose length is equal to $|X| = N$. It drops all shorter traces and, for longer traces, considers all contiguous sub-traces of length $N$. Then compute the distance from every candidate trace to the NSS of $X$. If this distance is under some threshold $\tau_N$ that depends on $N$, declare a successful match. Figure 5 illustrates how the trace "matches" the fingerprint of the correct text but not those of other texts.

***Choosing $\tau_N$ to ensure no false positives.*** Let $\mathbb{V} \subseteq \mathbb{S}$ the set of texts whose NSS is variable, $X \in \mathbb{V}$ a variable text, and $Y \in \mathbb{S}$ any text s.t. $X, Y$ have the same length $N$ but are not similar. Let $t$ be a trace measured while the user was typing $Y$. We want to avoid false positives, i.e., $Y$'s trace mistakenly matched to $X$: $\|\mathcal{I}_{\mathcal{M},q}(X) - t\| < \tau_N$. Let $T^Y$ be the probability distribution of traces measured while the user is typing $Y$, and let $d(N)$ be the bound on the attacker's measurement error:

$$Pr_{t \leftarrow T^Y}[\|\mathcal{I}_{\mathcal{M},q}(Y) - t\| < d(N)] \geq 1 - \epsilon \quad \text{for some small } \epsilon \tag{1}$$

From Section 3.2, we have that, for uniformly sampled $X$ and $Y$,

$$Pr_{X,Y \xleftarrow{U} \mathbb{V} \times \mathbb{S}}[\|\mathcal{I}_{\mathcal{M},q}(Y) - \mathcal{I}_{\mathcal{M},q}(X)\| > U(N)] \geq 1 - \epsilon \tag{2}$$

For any $t, X, Y$ such that the events in Equations 1 and 2 hold, the distance from $t$ to $X$'s fingerprint is bound by the triangle inequality: $\|\mathcal{I}_{\mathcal{M},q}(X) - t\| \geq U(N) - d(N)$—see Figure 4. By setting the threshold $\tau_N \leftarrow U(N) - d(N)$, we guarantee that for random $X \in \mathbb{V}$ and $Y \in \mathbb{S}$, the probability of a false positive where $\|\mathcal{I}_{\mathcal{M},q}(X) - t\| < \tau_N$ is at most $2\epsilon$ (by union bound).

---

**Algorithm 2** Matching a sequence to trace

1: **procedure** MATCHING($X, candidates$)
2:    $\mathcal{I}_{\mathcal{M},q}(X) \leftarrow$ FIND_NSS($\mathcal{M}, q, X$)
3:    **if** $\mathcal{I}_{\mathcal{M},q}(X)$ not *variable* **then**
4:       **return** *not_variable*
5:    **for** $t \in$ GENTRACES($candidates, |X|$) **do**
6:       **if** $\|\mathcal{I}_{\mathcal{M},q}(X) - t\| < \tau_{|t|}$ **then**
7:          **return** $t$
8:    **return** *no_match*

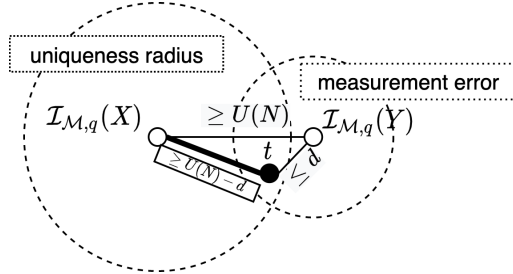

Figure 4: Bound on measurement error and uniqueness radius ensures no false positives.

***Beyond distance-based matching.*** Our matching algorithm is simple, very conservative, and amenable to theoretical analysis. A real-world attacker who is not interested in provable guarantees could use much more sophisticated methods. Convolutional neural networks often outperform distance-based methods [38, 40], especially with noisy measurements [38]. For our task, these methods are likely to be effective even when using a very noisy side channel where $d(N)$ is higher than $U(N)$. Empirically demonstrating their precision for time-series fingerprint matching with an extremely low base rate [4] would take many billions of measurements, however.

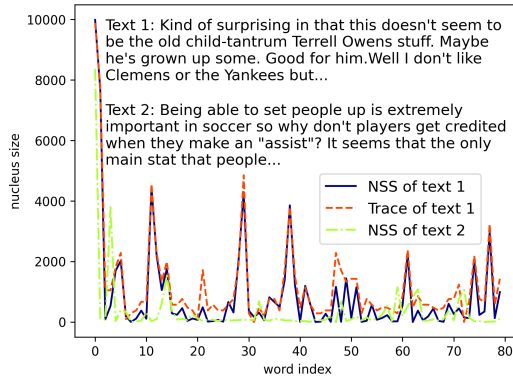

Figure 5: The trace matches its text's NSS, and does not match to another text's NSS.

# 5 Fingerprinting via a cache side channel

Our proof of concept uses a Flush+Reload attack (see Section 2.2). For this attack, we show that the noise $d(N)$ of the attacker's measurements of nucleus sizes is much smaller than the uniqueness radius $U(N)$ of nucleus size series. This attack thus has high recall and **no false positives** (see Section 4.2). We also demonstrate that **uniqueness radius grows faster than measurement noise** as a function of the sequence length $N$. Therefore, even for noisier measurements from a different side channel, machine, or software setup, we expect that there exists an $N$ such that $U(N) >> d(N)$.

## 5.1 Experimental setup

Our "victim" uses an auto-completion app based on Hugging Face's PyTorch code driving a GPT-2-small language model, as in Section 3.2. We used Hugging Face [23] and Pytorch [34] code versions from, respectively, 7/18/2019 and 7/22/2019. The victim and attacker run as (isolated) processes on the same core of an 8-core, Intel Xeon E5-1660 v4 CPU. If PyTorch is installed on the machine, the `libtorch.so` shared object (SO) is in a public, world-readable directory. The victim loads this SO into their process. The attacker loads the same SO, thus the SO's physical memory addresses are mapped into the virtual process space of both the attacker and the victim (operating systems have a copy-on-write policy for SOs in physical memory). The attacker uses Flush+Reload to monitor the first instruction of a function called within the loop, as shown in Figure 6.

```
159cfb0:        48 89 de            mov     %rbx,%rsi
159cfb3:        4c 89 ef            mov     %r13,%rdi
159cfb6:        e8 e5 9f ff ff      callq   1596fa0 <_ZN2at6native12_GLOBAL__N_17Indexer3getEl.constprop.201>
                .
                .
159cfdc:        75 d2               jne     159cfb0
```

Figure 6: Assembly code of the loop in lines 14-15 of Algorithm 1, implemented in `libtorch.so`.

To determine when typing starts or ends, the attacker can use any side channel from Section 2.2 to probe the auto-completion application or shared libraries. For segmenting the trace into prefixes, the attacker can use CPU timestamps for each Flush+Reload hit to identify the gaps. Measured traces must be processed to remove noise and outliers—see Appendix A in supplementary materials.

Human editing such as deleting or rewriting "pollutes" the measured trace with nucleus sizes corresponding to deleted subsequences. This may cause false negatives, but not false positives. The attacker can try to guess which trace chunks originate from edits and remove them before matching. When there are many edits, this has a nontrivial computational cost, which can be offset by using auxiliary information from the side channel to guide the guesses (e.g., timing, nucleus sizes, and control flow of the code that operates the language model). We did not evaluate these methods because human editing is difficult to simulate at scale, and leave them for future work.

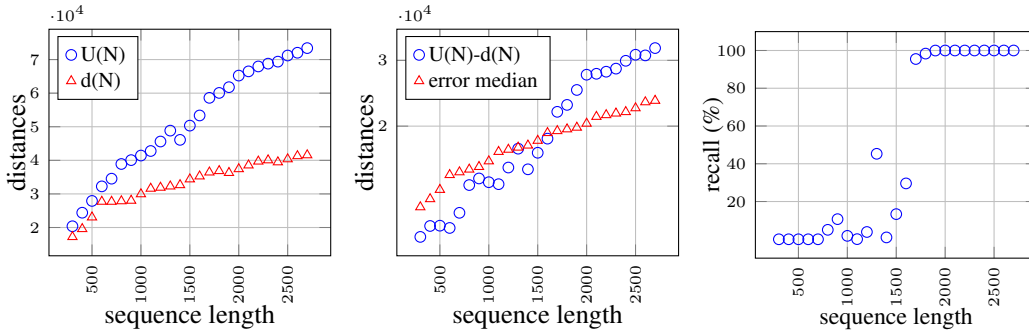

(a) Uniqueness radius vs. upper bound of measurement error.

(b) Gap between uniqueness radius and error vs. median error.

(c) Attack recall for non-noisy traces (after processing).

Figure 7: Measurement error and attack recall.

## 5.2 Measurement error and attack recall

The analysis in Section 4.2 assumes that the side-channel measurement error is bounded by some $d(N)$. After measuring 1566 traces from the reddit-sports dataset and removing noisy traces, we fit a

normal distribution and set $d(N)$ to 10 standard deviations above the mean. The fit is tight but not perfect—see Appendix A. Figure 7a shows $d(N)$ as a function of $N$, and how the uniqueness radius $U(N)$ diverges from $d(N)$.

This illustrates the fundamental characteristic that enables our attack: **pairwise distances between fingerprints grow faster as a function of sequence length $N$ than the attacker's measurement error**. Therefore, if a sequence is long enough, the attacker can match the fingerprint without false positives. This property can hold for any side channel, not just Flush+Reload. The key requirement is that the (squared) error of a single measurement is, on average, smaller than the (squared) difference between the nucleus sizes in the same position of different sequences.

In Algorithm 2, we set the threshold $\tau_N$ to $U(N) - d(N)$, so that the recall, after noisy trace filtering, is equal to the probability that measurement error is below $U(N) - d(N)$. Figure 7c shows recall for different $N$: when $N \geq 1900$, recall is greater than 99%. When accounting for the 6% of traces that were filtered out as too noisy (Appendix A), this is equivalent to recall >93%.

### 5.3 Case studies

We show that users of real-world anonymous forums would have been vulnerable if they had used auto-completion based on nucleus sampling.

***Silk Road forum.*** We used an archive of Silk Road forum posts [39] created by one of the participants in October 2013, after the Silk Road marketplace was shut down but before the forums were taken offline. For each of 41 users who had at least 2700 words in their posts, we concatenated their posts in chronological order into a single sequence and generated the corresponding NSS fingerprints. An individual post has 50.6 words on average.

We simulated the auto-completion process for each user's sequence using the Hugging Face Transformers language generator and applied our proof-of-concept attack from Section 3.2. In reality, posts may be separated by unrelated typing, but (a) it is relatively straightforward to identify the current application via techniques from Section 2.2, and (b) the attacker knows when the typing begins and ends (see Section 5.1). To ensure even stronger isolation, we ran the attack process in an AppArmor [3] sandbox (by default, it still lets the attacker read PyTorch shared objects). These experiments were done on the same machine as in Section 5.1.

We truncated all traces to $N$=2700 and filtered out NSS that are not sufficiently variable. This left 18 users out of 41. For each of them, we computed the measurement error of the attack, i.e., the distance between the measured trace and NSS—see Appendix B in supplementary materials. In all cases, the error is less than $U(N) - d(N)$, thus the attack would have been able to correctly de-anonymize these 18 users with no false positives or false negatives.

***Ubuntu Chat.*** We selected the 200 most active users from the Ubuntu Chat corpus [41] and followed the same procedure as above to generate the NSS fingerprints of their posts. The average post length is 9.5 words. Because posts are short, the sequences of all selected users are sufficiently variable. Filtering out noisy traces and (for technical reasons) 2 users with irregular characters in their usernames left 186 traces. For all of them, the error was less than $d(N)$, so there were no false positives, and for all except one, the error was less than $U(N) - d(N)$, so they were identified correctly. The overall recall is 93.4%.

## 6 Mitigation

To replace Algorithm 1, we suggest Algorithm 3 which follows two standard guidelines for cryptographic code. First, it avoids data-dependent memory accesses at a granularity coarser than a cache line [6, 16], thus an attacker cannot mount a Flush+Reload attack to count how many times a code line executes. Whereas Algorithm 1 iterates over indices $i$ where $cum\_probs[i] > p$ (testing if $i$ is within the $p$-nucleus) and assigns $-\infty$, Algorithm 3 entirely avoids control flows that depend on the condition $cum\_probs[i] > p$.[1] Second, execution time should not be correlated with secret data [11]. Figures 8a and 8b show the relationship between the nucleus size and execution time of the token

removal loop with and without the mitigation, indicating that our implementation (which has a fixed number of iterations) reduces the correlation.

The cost of our mitigation is a 1.15x average slowdown in the loop execution time, which translates into only a 0.1% increase in the runtime of SAMPLE_SEQUENCE (which itself accounts for a tiny fraction of the execution time relative to the encoder/decoder passes). When simulating auto-completion on a 2700-word sequence in the setup from Section 5.1, there was a negligible, 0.3% runtime difference in favor of our algorithm, implying that the difference between Algorithms 1 and 3 is dominated by other factors, such as natural fluctuations in the CPU load.

**Algorithm 3** Top-p filtering with a fixed number of loop iterations.

```
1: procedure TOP_P_FILTERING(logits, p)
2:     sorted_logits, indices ←ARGSORT_DESCEND(logits)
3:     cum_probs ←CUM_SUM(SOFTMAX(sorted_logits))
4:
5:     for i ∈ {1...LEN(logits)} do
6:         z ← FLOAT(cum_probs[i] > p) · MAXFLOAT · 2
7:         logits[indices[i]] ← logits[indices[i]] − z
8:
9:     return logits
```

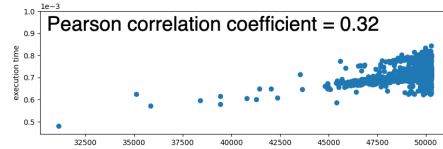

(a) Nucleus size vs. run time (original)

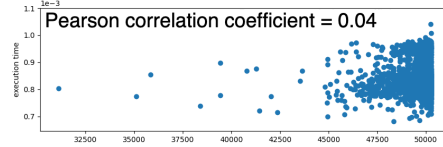

(b) Nucleus size vs. run time (mitigation)

Figure 8: Execution time of the token removal loop with and without the mitigation.

No implementation is immune to side channels, however. Address bits within a cache line could still leak through the cache on certain processors [48]. Even without input-dependent paths, loop runtimes may still slightly depend on the input due to value-dependent execution times of floating-point operations [2] (an attacker must be able to measure the loop very accurately to exploit this). Mitigations of these and other side-channel risks incur implementational and runtime overheads [11].

We believe that our implementation strikes a good balance by substantially increasing the gap between what side-channel attacks can achieve on specific platforms in controlled laboratory conditions vs. what is available to real-world attackers. We argue that removing "easy" targets like input-dependent loops should be a minimal security standard for core ML building blocks.

# 7 Related work

Prior work showed how to infer model architectures and weights—but not inputs—via model execution time [9], addresses of memory accesses leaked by GPUs [21] and trusted hardware enclaves [22], and or via cache [20, 46] and GPU [30] side channels.

The only prior work on inferring model inputs required hardware attacks, such as physically probing the power consumption of an FPGA accelerator [42], physically probing an external microcontroller executing the model [5], or inferring coarse information about the input's class from hardware performance counters [1]. To the best of our knowledge, ours is the first work to show the feasibility of inferring neural-network inputs in a conventional, software-only setting, where the attacker is limited to executing an isolated malicious application on the victim's machine.

# 8 Conclusions

We used nucleus sampling, a popular approach for text generation, as a case study of ML systems that unwittingly leak their confidential inputs. As our main technical contribution, we demonstrated that the series of nucleus sizes associated with an English-language word sequence is a fingerprint which uniquely identifies this sequence. We showed how a side-channel attacker can measure these fingerprints and use them to de-anonymize anonymous text. Finally, we explained how to mitigate this leak by reducing input-dependent control flows in the implementations of ML systems.

## Broader Impact

This work will help improve security of ML code by (a) identifying a new category of potential vulnerabilities faced by ML systems that operate on sensitive data, and (b) explaining how to design implementations so as to mitigate this risk. This research will primarily benefit implementors of ML models and, in general, increase trust in ML systems.

## Acknowledgments and Funding

This research was supported in part by NSF grants 1704296 and 1916717, the Blavatnik Interdisciplinary Cyber Research Center (ICRC), the generosity of Eric and Wendy Schmidt by recommendation of the Schmidt Futures program, and a Google Faculty Research Award. Roei Schuster is a member of the Check Point Institute of Information Security.

## Footnotes

[1]To this end, we use the expression FLOAT($cum\_probs[i] > p$) · MAX_FLOAT · 2 in Line 6, which resolves to either $\infty$ if $cum\_probs[i] > p$, or 0 otherwise. The multiplication by 2 invokes a float overflow in the case of $cum\_probs[i] > p$, which resolves to $\infty$, as per IEEE 754 floating point arithmetic standard [26].

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
