[Supplementary Material]

# A   Trace preprocessing

***Challenge: loop iterations are faster than Flush+Reload.***   The attacker's goal is to infer the number of iterations of the token removal loop (see Section 3.3). In the Reload phase of the Flush+Reload attack, the attacker learns whether the victim has accessed an address since it has been Flushed (see Section 2.2). A naive attack would iteratively perform Flush+Reload and receive indications whenever the victim accesses this address, which happens on every iteration of the target loop.

(a) Measurement error of 1566 2700-word traces

(b) Part of a noisy trace; red circles indicate outliers

(c) Measured number of iterations vs. CPU cycles: normal trace

(d) Measured number of iterations vs. CPU cycles: noisy trace

(e) Noise level vs. distance to fingerprint

(f) Measurement error after removing noisy traces

Figure 9: Filtering out noisy traces.

The problem is that Flushing and Reloading is two orders of magnitude slower than executing the target loop. If the victim performs more than one iteration per each attacker iteration, the attacker misses accesses. In our environment, the naive approach only captures a small fraction of the victim's iterations. Fortunately, we observe that the fraction of the victim's iterations captured by the attacker is consistently around 1.1%. Therefore, to estimate the actual number of iterations, the attacker can simply multiply the measured number by $100/1.1$.

***Challenge: some traces are very noisy.*** Figure 9a shows the distribution of measurement error over 1566 2700-word traces from the reddit-sports dataset. The distribution has a long tail due to several outliers where the error is very high. Figure 9b indicates that outliers are associated with periods when the Flush+Reload loop was slower or produced more false negatives (an address access by the victim was not indicated by a lower load time). This can be due to activity by concurrent processes sharing the attacker's core, load on the cache bus, or other low-level interactions.

***Filtering out noisy traces.*** We observe that in a normal state, the execution time of each iteration of the target loop is usually close to a constant. Figure 9c shows the relationship between the measured number of iterations and time (in CPU cycles). There are some outliers (likely caused by CPU interrupts), but the relationship is almost linear. If a trace is noisy, however, the correlation is weaker—see Figure 9d.

We measure the "noise level" of a trace as the mean squared distance of its (iterations, time) series relative to the expected line. Figure 9e shows the relationship between the noise level of a trace and the distance to its corresponding fingerprint. For our experiments, we removed the 6% of the traces with the highest noise levels. Figure 9f shows the histogram of measurement error after removing these traces. This histogram fits a normal distribution model, with symmetry and exponential decay, except for a few outliers where the measurement error is several standard deviations away from the mean, indicating that the fit is imperfect. Even so, the error is always far below $d(N)$, so these outlier traces would not cause the matching algorithm to produce a false positive. For higher values of $N$, we expect to also avoid a drop in recall because the uniqueness radius increases faster than the measurement error (Figure 7b).

## B   Data for the case studies

Table 1 shows variability and measurement error of the nucleus size series corresponding to the posts of Silk Road forum users.

| user | variability | measurement error |
|---|---|---|
| cirrus | 1546.78 | 25635.48 |
| indica9 | 1763.87 | 22261.53 |
| stealth | 1455.91 | 24129.27 |
| cirrus(SR2) | 1753.64 | 20955.76 |
| jediknight | 1465.62 | 22010.80 |
| digitalalch | 1586.34 | 25537.31 |
| synergy | 1699.32 | 22907.05 |
| ssbd | 1701.14 | 23879.66 |
| captainwhitebeard | 2004.27 | 20950.86 |
| colorblack | 1502.06 | 20909.17 |
| nomad bloodbath | 2032.92 | 23724.85 |
| envious | 1504.53 | 22530.60 |
| modziw | 1737.35 | 21170.16 |
| v | 1586.50 | 21646.63 |
| sarge | 1522.02 | 20902.54 |
| warweed | 1667.98 | 22507.60 |
| samesamebutdifferent | 1521.18 | 21508.18 |
| scout | 1521.50 | 22985.98 |

Table 1: Variability and measurement error for Silk Road Forum users. ($N$ is 2700, $U(N) - d(N)$ is 31860).

Figure 10 shows the distribution of the measurement error of the nucleus size series corresponding to the posts of the 200 most active Ubuntu Chat users. All posts are sufficiently variable.

Figure 10: Measurement error for Ubuntu Chat users. ($N$ is 2700, $U(N) - d(N)$ is 31860, $d(N)$ is 41551).