[Reviews · NeurIPS 2020]

Review 1

Summary and Contributions: This paper examines the security of a specific sampling algorithm—Nucleus Sampling—to see if it can be used to de-anonymize autocompleted text from a specific user. First the authors layout the basic groundwork of generation and microarchitectural side channels. Next they formalize the notion of fingerprint and conjecture that the Nucleus Size Series is a fingerprint for long enough text sequences that are sufficiently variable. Next, a threat model is introduced that takes advantage of the fact that the most common implementations of nucleus sampling leaks the nucleus sizes through side channels, using side channel information and offline model re-running of anonymous text to deanonymize users. The authors then go through how to enact this attack with a cache side channel and simulate a case study of de-anonymizing a deep web forum. A mitigation solution is proposed, which results in a less than 1% slowdown, but safer execution. Finally, implications and broader impact is discussed, mostly regarding how to make ML algorithms safer. I have read the rebuttal.

Strengths: . This paper shows how a popular approach to text generation can be used to de-anonymize text 2. Both a formal system that is easy to mathematically analyze and a real case study are shown, which support each other nicely 3. The threat model is well explained and the initial solution is cheap enough to be of practical value 4. The paper is well-written, with helpful diagrams, good explanation of terminology, and some key insights, e.g. that the distance between fingerprints grows faster as a function of sequence length than measurement errors.

Weaknesses: 1. Some of the assumptions made will not hold in the real-world. 2. The real-world case study was useful, but more than one domain of text and testing the potential solution would have made this paper more rigorous.

Correctness: A few assumptions in this paper are unlikely to hold in real life: 1. Being aware of the _exact_ model that was used to produce the given text. Does this hold for models that have been finetuned? Does it hold for identical models trained on the same data with two different random seeds? Does it hold for models that are merely similar in size and data? -> These things take significantly more resources to investigate, but at least looking at finetuning which is quick and built-in to most libraries (including HuggingFace, which the authors use) would be key to making these results actually realistic. That said, currently people _do_ tend to use exactly the same model in many cases, i.e. GPT-3 which cannot be easily finetuned, so I will not harp on this point too much. 2. This paper assumes that the NSS from a user will be the same if they (a) autocomplete an entire document or (b) accept intermittent autocompletions over time. It also assumes no editing on the part of the user. -> These are both hard to simulate, as they involve real people and their choices in the real world would differ from a laboratory setting, but at least acknowledging and discussing this issue is a must. Simulation using other algorithms in-between Nucleus Sampling would also be a nice middleground. 3. NSS contain larger terms the less text they are conditioned on. This information could skew the analysis done here. -> I believe this is beyond the scope of an initial paper to _test_, but again some discussion of how this would play out in the real world seems necessary.

Clarity: I find this paper extremely well written. The explanations are clear and, at least to my competency, appear rigorous. The diagrams are extremely helpful and understandable.The introduction to security concepts will make this accessible to many at NeurIPS.

Relation to Prior Work: Yes, as far as I know, but I am not well-read in the security literature.

Reproducibility: Yes

Additional Feedback: I enjoyed reading this paper quite a bit, and the only thing stopping me from giving it a higher score is the fact that the case study is so limited. Three things that can be done to increase its effectiveness are: 1. Test the solution given on the same dataset that was tried for the original algorithm and see if it actually helps. 2. Try the attack on another dataset. 3. Try the attack on the same or another dataset where many small texts are pieced together. 4. Try the attack on the same or another dataset with the original language model, but text generated by a language model finetuned on different users’ posts. If the authors agree to add any two of the above, I am willing to raise my score, unless another reviewer notes a significant flaw I have missed. --- Line 196: You never introduced \tau_N Even though it’s clear from context, it should be introduced for the benefit of the reader who is not familiar with this area._


Review 2

Summary and Contributions: This paper use nucleus sampling as a window to study how to unanonymize anonymized text input. I find the algorithm straight forward and results interesting. The topic is novel and the impact in the future study can be huge

Strengths: An interesting direction that is novel and may have huge impact in the future.

Weaknesses: It would be great if the paper can give some intituive textual example to walk the reader through the algorithm

Correctness: To my knowledge, the experiments look correct

Clarity: The paper is well written

Relation to Prior Work: This is a new direction, very few related work are present.

Reproducibility: Yes

Additional Feedback:


Review 3

Summary and Contributions: The paper shows that text prediction ML models are susceptible to side-channel attacks. It uses the size of the candidate set of tokens that are predicted after each word to generate a "fingerprint" for a sequence. The paper claims and empirically validates that for long enough sequences this fingerprint is "unique" to a sequence. As a result if an attacker can obtain a fingerprint of a user typed sequence it can then generate sequences itself and find those that are "close" to the one typed by the user. As a side-channel the authors use a cache (e.g., a scenario where an attacker runs a process on the same machine as a text prediction software and loads same shared libraries) and a known attack (Flush+reload) to understand which part of the code the text generation process (C) is executing (it does so by flushing the instructions and checking if they have been loaded since flushing: this can be done by measuring the time between flushing and reloading a particular instruction, since cache misses are slower than cache hits one can distinguish whether an instruction was loaded or not). Since ML code makes data dependent accesses, the attack is possible (i.e., loading one instruction vs. another reveals where in the code C is. The main scenario of the attack is de-anonymization of a user. The authors show that if they apply known techniques for mitigating attacks (e.g., removing "if" conditions) then attack can be protected against.

Strengths: - privacy of machine learning systems is an important topic - the paper thoroughly investigates side-channel leakage of a text generation model

Weaknesses: - the cache side-channel attack used in the paper is known - any code that makes data-dependent accesses will be prone to side-channel attacks, machine learning code included; this is also not that surprising - the defense is an application of well-known techniques - sequence length has to be quite long for a successful attack ( > 1500 words ), might not be a realistic setting

Correctness: Yes. The paper establishes a very clear methodology for evaluating its claims and presents empirical evidence to validate them.

Clarity: Yes, this is the strongest part of the paper. It is clearly presented.

Relation to Prior Work: This is probably the weakest part of the paper as it combines a known side-channel attack with a new ML algorithm. Code that has data dependent accesses is known to be prone to these attacks and this was demonstrated for a variety of applications (including application working with text: keystrokes [27], spell checking*). *Controlled-Channel Attacks: Deterministic Side Channels for Untrusted Operating Systems by Xu et al S&P'15.

Reproducibility: Yes

Additional Feedback:


Review 4

Summary and Contributions: This paper presents a side channel attack on an text auto-complete system to enable de-anonymization of the text. It argues that artifacts from nuclear sampling (the nuclear sampling series, NSS) for text auto-completion can be used to fingerprint a large number of English word sequences. In the proposed threat model, the attacker runs in a sandboxed process on the same OS as where the user's text auto-complete app runs and is assumed to have complete knowledge of the ML model used for auto-complete. By doing flush+reload cache side channel attacks, the attacker can infer the NSS (the fingerprint) for the text being entered. The attacker can then check this fingerprint against the fingerprint of anonymous posts, such as those on Reddit, to de-anonymize them. Experiments show that this attack can often be successful. The authors raise awareness about side-channel attacks on ML through this work. I have read the rebuttal and generally agree with it.

Strengths: - A novel attack on an ML system showing the risk on side channel attacks. - Demonstrate that nucleus sampling, in its usual implementation, leaks user-typed text. - Propose implementation-level mitigations that defeat side channel attacks by removing relationship between the nucleus size and the number of iterations.

Weaknesses: - That a fingerprint is unique to a text sequence is not proven but only conjectured. However, the paper does present experiments that suggest uniqueness can be presumed for practical purposes. - The proposed reliance of loading the same shared objects appears quite fragile. The user can easily defend against this attack by loading their own copy of the pytorch so. An attacker may use prime+probe but the effectiveness of this attack has not been demonstrated here. - Ultimately, text auto-complete seems like a non-essential application unlike cryptography. That is, if a user were concerned about side channel attacks, they could simply avoid using auto-complete but they cannot avoid using cryptography. It may have been better if the author had attacked some application of ML, which the user could not avoid.

Correctness: Yes.

Clarity: Yes.

Relation to Prior Work: Yes.

Reproducibility: Yes

Additional Feedback:

[Author Response · NeurIPS 2020]

We thank the reviewers for their insightful feedback.

***Reviewer 1*** **Knowledge of the exact model and weights used by the victim:** Model reuse is indeed commonplace. If the victim is using an off-the-shelf autocompletion app (e.g., as part of a commercial software package), the same app is likely available to many other users, including potential attackers. We will clarify this in Section 4.1. Transferability of fingerprints across similar (but not identical) models is an interesting topic for future research.

**Fully generated texts vs. intermittently accepting a completion**: For a given text, this does not affect the trace measured by the attacker because in both cases the nucleus is sampled after every word (we will clarify this in the text). Human editing such as deleting or rewriting might "pollute" the attacker's trace with nucleus sizes corresponding to deleted subsequences. The attacker can deal with this by guessing which trace chunks originate from edits, removing them, and trying to match the result. When there are many edits, this has a nontrivial computational cost, which may be offset by using auxiliary information from the side channel to guide the guesses (e.g., timing, nucleus sizes, and control flow of the code that operates the language model). Such methods are hard to evaluate in our lab setting because human editing is indeed hard to simulate. We will expand the discussion of this limitation in Section 5.1.

**Conditioning NSS on longer texts**: In our experiments, the language model resets on every user post (sometimes referred to as "sentence" in the submission; we will fix the terminology). We will expand on the implications of longer posts in Section 3.2, providing the average post lengths for the 5 subreddits in our experiments (they range from 36 to 61 tokens). Average post length is inversely correlated with the fraction of "variable" sequences (variability ensures that the NSS is a unique fingerprint), but even for the real-world text domains with relatively long posts (e.g., sports), the fraction of variable sequences is significant (>40%). We acknowledge that for NSS generated from much longer individual texts (e.g., entire documents with thousands of tokens), the fraction of variable sequences may be lower.

**Additional experiments:** We thank the reviewer for their suggestions. (1) We note that our mitigation makes loop iteration counts independent of the nucleus sizes, therefore the attack fails completely. We will explain that it would even fail to distinguish 2 known texts, let alone recognize an open-world fingerprint. (2) We will add another case study, similar to the one in Section 5.3. For example, we already collected traces of 50 users' aggregated posts in the Ubuntu Chat Corpus. The texts are all variable, their NSS are unique (distance $> U(N)$), traces are consistent with fingerprints (noise $< d(N)$), and our attack identifies them.

***Reviewer 2.*** As an additional illustration, we plan to add a plot similar to the one on the right, showing the NSS of two text sequences $X$ and $Y$ and the trace $t$ of $Y$. This illustrates how the trace "matches" the fingerprint of the correct text and does not match those of other texts.

***Reviewer 3.*** Side channels due to input-dependent branching have been studied in the computer security literature, yet none of the previous work (including none of the work cited by the reviewer) identified *inputs of ML models* as being vulnerable to these attacks. This observation and our analysis are nontrivial, because computation in ML models involves mostly tensor arithmetics whose control flow does *not* depend on the input. Yet we demonstrate that even a few lines of code in a widely used ML building block can unintentionally leak the inputs.

Our main contributions are to show that (1) text generation using language models creates fingerprints for many input texts, and (2) these fingerprints are measurable via side channels. Uniqueness of fingerprints increases with the input text's length *regardless of the specific side channel used*.

Our work neither requires, nor claims any innovation in side-channel techniques or defenses. We view the fact that nucleus size sequences can be fingerprinted via known techniques as a strength of our paper. It means that the vulnerability we identify is dangerous because it can be exploited using well-known, easily available tools and does not require esoteric or new ones.

***Reviewer 4.*** The reviewer correctly notes that loading a local shared object would thwart a Flush+Reload attack. That said, other side channels such as Prime+Probe have been repeatedly shown effective for inferring loop arguments (Section 2.2).

Users can indeed try to avoid using autocompletion features altogether, which is not always the case for cryptographic primitives. Our work helps users make these decisions by identifying nucleus sampling (and ML building blocks in general) as security hotspots, analogous to cryptographic primitives in that they are small functional components that are (1) ubiquitously deployed and integrated into many software apps, and (2) operate on sensitive data. This motivates adoption of secure coding practices for ML code.

[Meta-Review · NeurIPS 2020]

This paper generated a significant amount of discussion. SCIENTIFIC: Regarding the purely scientific aspects, the reviewers discussed about the novelty of the contribution. On the one hand, if one takes the point of view of the security community, the proposed attack and defense are known and the vulnerability is not surprising since any data-dependent accesses is prone to side-channel attacks. On the other hand, from the point of view of the machine learning community where these concerns are currently not well known, the paper presents very clearly a reasonable approach to start thinking about security of machine learning and NLP code using actual algorithms that text generation researchers and practitioners use. The paper can thus serve a useful cross-discipline discussion. In the end, there was a consensus to say that the latter aspect outweighs the former. ETHICS: The paper was also flagged by one reviewer as raising potential ethical concerns due to the use of data scrapped from the infamous Silk Road forum and (to a lesser extent) the lack of clear policy related to responsible vulnerabilities disclosure communication towards Hugging Face. The paper was therefore sent to three ethics experts for review, which then generated further discussion among reviewers, myself and the senior area chair. Many perspectives were considered regarding the issues such as (quoting the ethics reviews): - data provenance: "The URL for the Silk Road data does not clarify the provenance much". - offensiveness of the content: "the archive dataset used for this case study contains inappropriate and offensive text content, detailing now known illegal activity and explicit racial/misogynistic slurs. But it was also recognized that many/all forum-like datasets such as comments in Fox News articles, Washington Post articles, and on Amazon would have on the order of .01-10% content one can find offensive. - setting a potentially dangerous precedent: "If we ban this dataset because it has some offensive content, will we ban all datasets that have any offensive content?" - appropriateness of the match between the choice of dataset and the goals of the research: the dataset was seen as a use-case where "individuals are truly speaking their mind and wish for anonymity", which is important for the purpose of the research. It was pointed out that other sources such as text from online health forums (e.g., https://zenodo.org/record/1479354) could also be appropriate although they may be more formal and not as individualized communication. DECISION AND ACTIONS FROM THE AUTHORS: In the end, although there were scientific/ethical pros and cons, the final decision is to accept this paper conditionally on the following actions being taken by the authors: 1/ In the final version of the paper, include an explicit note warning people that the dataset has offensive material, and that their use of it is because it is representative of anonymous informal communications, and that the data should not be construed to represent the opinions of the authors. 2/ Notify Hugging Face of the vulnerability before publication. Following the suggestions of R1, we also recommend the authors to add a further case study to assess the influence of the sequence length. ******************************* Note from Program Chairs: The camera-ready version of this paper has been reviewed with regard to the conditions listed above, and this paper is now fully accepted for publication.